# A propensity index for surface runoff on a karst plateau

Christian Reszler[1], Jürgen Komma[2], Hermann Stadler[1], Elmar Strobl[3], Günter Blöschl[2]

[1]JR-AquaConSol GmbH, Steyrergasse 21, 8010 Graz, Austria
[2]Vienna University of Technology, Institute for Hydraulic and Water Resources Engineering, and Centre for Water Resource Systems, Karlsplatz 13, 1040 Vienna, Austria
[3]Technisches Büro – Ingenieurbüro Mag. Dr. Elmar Strobl, Grubbergweg 4, 8062 Kumberg

*Correspondence to*: Christian Reszler (christian.reszler@jr-aquaconsol.at)

**Abstract.** Karst aquifers are an important water resource, but are particularly vulnerable to pollution due to the typically short residence times. As the rainwater runs off on the surface it may collect contamination from faeces and other sources, before infiltrating. It is therefore important to understand the spatial distribution of the frequency of surface runoff in Karst areas. This paper proposes a new field mapping method for the ability of the landscape to produce and convey surface runoff. The mapping method is based on (i) prior spatial information (e.g. geological map, terrain model), (ii) a visual assessment from a distance at the landscape scale (e.g. traces of surface runoff) and (iii) local data collection in the field (e.g. soil moisture, grain size distribution). The focus on variables that can be assessed from a distance in the landscape makes the method suitable for mapping larger areas than traditional field mapping. The mapping method is developed and tested for the 60 km² Hochschwab area in Austria. The field mapping is used to specify a surface runoff propensity index which is tested against the spatial distribution of observed sink holes in the area. The mapping indicates that, in the study region, runoff occurs much more frequently in the poorly karstified dolomitic areas than in the limestone areas that are highly karstified. On dolomites, low permeable soils or debris prevail, often resulting in a permanent surface drainage network. On karstified limestone, sometimes overlaid by debris, surface runoff only occurs through infiltration excess at high rainfall intensities. Overall the analyses suggest that the mapping method is suitable for efficiently and reliably identifying spatial patterns of the ability of the landscape to produce and convey surface runoff in karst areas.

# 1 Introduction

Karst groundwater is an important source for drinking water in many countries around the world. In Austria, numerous karst springs exist, e.g., in the Northern Calcareous Alps, where the water is used for the drinking water supply of Vienna (e.g., Plan et al., 2010). In karst areas, soils tend to be shallow, and the fast drainage through the karst system make karst springs very vulnerable to pollution in the source areas. Pollution may include faeces from cattle farming and wild animals, tourism and accidental fuel spills on forest roads. Safeguarding the quality of karst springs therefore involves land management of the source areas in order to minimize the risk of pollution. An understanding of the processes of surface runoff may enhance

the efficiency of the land management by quantifying the hazard of contaminant input into the karst system, particularly during high intensity, convective storms. During convective storms, flow processes tend to accelerate (Bonacci, 2001a), they therefore play an important role for material mobilization and transport. The longer contaminants are in contact with the surface water, the more likely is their transport to sink holes and further into the karst system towards the spring. Frequent

surface runoff in karst areas can therefore be associated with high contamination risk, all other factors being equal.

At the plot scale, the propensity of areas to produce surface runoff is usually mapped by field surveys (e.g., Peschke et al., 1998; Scherrer and Naef, 2003) based on a detailed mapping of hydrological properties, often including sprinkling experiments (Markart and Kohl, 2004). Such mapping exercises have been used to develop methodologies for identifying Hydrological Response Units (HRUs). HRUs are based on the idea that areas of similar physiographic properties, such as

slope aspect, vegetation type and soil type, imply hydrological similarity, i.e. similar runoff response (Flügel 1995). The way the layers of information are combined can have various degrees of process representation. For example, Flügel (1995) combined the layers by reasoning for a unit to represent 'rangeland on gley soil at the valley floor with shallow groundwater over impervious bedrock'.

In karstic catchments, traditional mapping methods tend to focus on the geology and the hydrogeology (e.g., Andreo et al.,

2006; Goldscheider, 2015), which comprise the lithology, the existence of faults and their permeability, the location of karst forms (e.g., dolines, poljes) and, less often, soil surveys. The focus on geology may be related to the wider availability of geological maps than soil maps and to the deep subsurface flow paths feeding karst springs. Sprinkling experiments in the limestone region of southeastern Spain were used for elucidating the mechanisms of runoff generation and infiltration. For example, Calvo-Cases et al. (2003) found that runoff on bare patches was mainly generated by the Hortonian mechanism. At

the slope scale, Hortonian discontinuous runoff occurred on the most degraded slopes or during high intensity rain events, and mixed runoff generation on less degraded slopes or on previously wet soils. For both types, there was a patchwork of runoff and runon areas. Li et al. (2011) found vegetation cover, litter, and rainfall intensity to be the main controls of infiltration rates, and the presence of bare soil and rainfall intensity to be the main controls of the runoff coefficient. Canton et al. (2016) found generally higher clay contents towards the foot of hillslopes which explained the increase of soil moisture

downslope at depths of 0.1–0.35 m. Macropores (rock and soil cracks) disconnected the surface water fluxes and reduced run-on to the lower part of the hillslope during wet periods. The spatial connectivity of flow processes tends to enhance fast contaminant transport (Western et al., 1998; 2001), but Fiener et al. (2011) point out that the effects of patchiness and the spatial organization of landscape characteristics on flow and transport processes are still poorly understood.

Fu et al. (2016) studied the role of "epikarst" in near-surface hydrological processes in a subtropical karst region of

southwestern China based on plot-scale rainfall simulation experiments. They found that more than 70% of the total rainfall moved vertically through the shallow soil layer and was redistributed as subsurface flow at the soil–epikarst interface or percolated deeper into the rock. Fu et al. (2015; 2016) and Chen et al. (2010) identified the spatial variability of the topsoil saturated hydraulic conductivity, surface soil water content and soil moisture dynamics as the main controls of infiltration in a small karst catchment in a subtropical region of China, and Peng and Wang (2012) found that surface runoff from the

karstic hillslopes in southwestern China was much less than that from non-karstic areas and that limestone fissures and fractures played important roles in surface runoff generation on karst limestone slopes due to their large infiltration and storage capacities.

At the scales of a couple of square kilometres or more that are more relevant from a management perspective, karst studies are traditionally based on GIS tools with little field validation. Vulnerability is often mapped by combining various layers of spatial data such as vegetation, land use, soil texture and geology (e.g., Flügel, 1995; Cost Action 620). Walter et al. (2000) presented a GIS based conceptual approach to assessing water quality risk in agricultural catchment based on the variable source area concept, following Frankenberger et al. (1999). They used the term hydrologically sensitive area (HSA) to refer to areas that are especially prone to generating runoff and therefore transporting contaminants to surface water bodies. HSAs are similar to HRUs, but differ from them in terms of their stronger emphasis on flow connectivity and transport of contaminants. The method was further developed by Thomas et al. (2016) by the use of LiDAR elevation models and by validation against rainfall and quickflow measurements. Similar to the HSA concept, Plan et al. (2009) identified "ultra-vulnerable areas" by a mapping procedure that involves the automatic delineation of local catchment boundaries of individual karst forms.

The purpose of this paper is to propose a method that allows the mapping of surface runoff propensity in karstic environments at the catchment scale. The main innovation of the method is that it allows the more rapid mapping of larger areas than the traditional geological mapping paradigm which consists of fully walking the region of interest. The study is set in a 63 km² karst plateau in Austria which is the recharge area of the main springs of the Vienna drinking water supply.

# 2 Case study area

The study area is part of the Hochschwab massif, one of the important karst areas in Austria. It consists of Triassic limestones and dolostones with a thickness of up to 2000 m. The karstified rocks sit on shales and sandstones of the Werfen formation (Mandl et al., 2002). The geological setup has led to intensive karstification including numerous cave systems and a range of surface karst forms such as dolines, karren, polja and ponors.

Due to the tectonic structure (Decker et al., 2006), the main karst spring (LKAS2) is situated at the northern slope of the Hochschwab massif. The catchment (Fig. 1) of the main spring covers an area of about 60 km². Elevations range from 650 to 2277 m a.s.l. with a mean elevation of 1720 m a.s.l. (Stadler and Strobl, 2006). The vegetation mainly comprises forests, krummholz-areas, alpine pastures on the plateau and badlands in the peak regions. Alpine farming and touristic activities during the summer months are important land use activities and potential sources for contamination of the karst aquifer. Additionally, faeces of wildlife may contaminate the aquifer.

A hydrogeological map was compiled in consistency with a detailed geological map (Mandl et al., 2002) and information from additional mapping surveys. The basic concept of compiling the map was the hydrogeological quality rating of the rocks, according to the "Rock Quality Designation (RQD)" concept of exploration geology (Stadler et al., 2016). This

hydrogeological quality rating describes fundamental properties of the rocks such as karstification capacity, tectonic stress related to water storage capacity and permeability, bedding, underline and fall. The objective was to combine different lithological units to units with similar hydrogeological behaviour.

The delineation of the catchment boundaries of the spring is based on two pieces of information: the detailed geological mapping and investigation of geological structures; and a long term analysis of stable isotopes, 18O and Deuterium, which provided robust estimates of the mean altitude of the spring (Stadler and Strobl, 2006). Spring discharge measurements with high temporal resolution (15 minutes) have been conducted by Vienna Water since 1992. A stream gauge at the Spitzboden site is used to assess whether the local stream is permanent or not. The catchment boundary differs significantly from the topographic boundary of the spring.

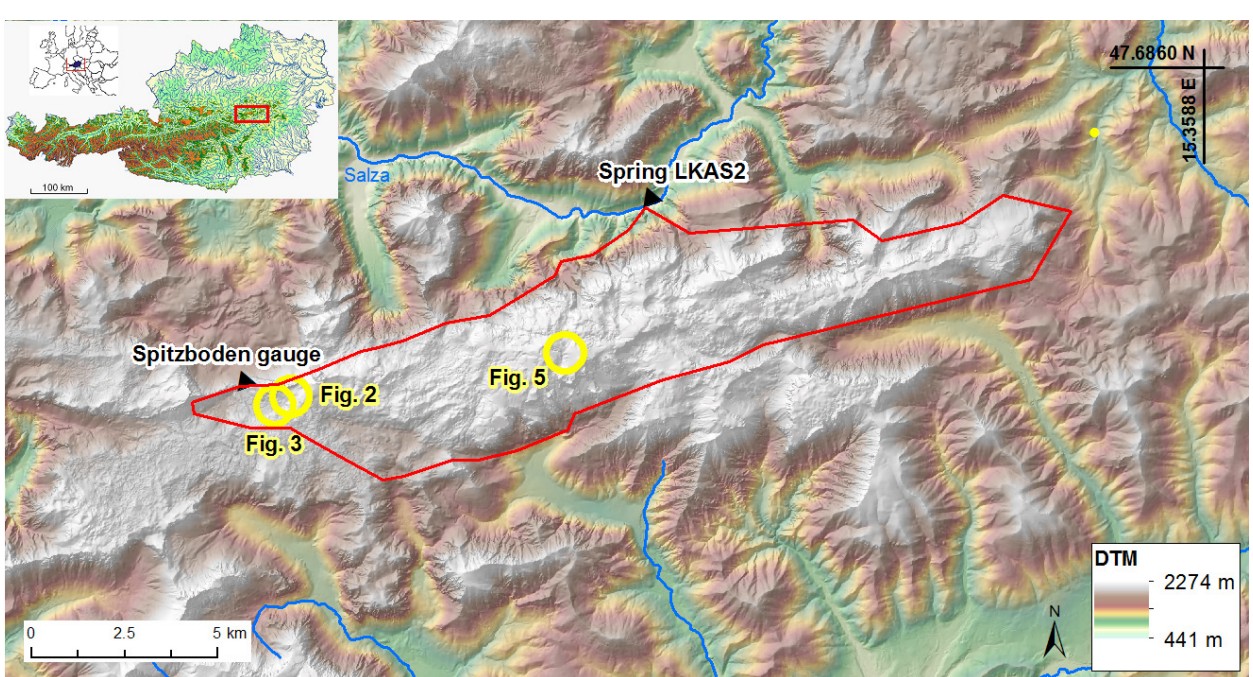

**Figure 1: Area subject to mapping (red line) in the catchment of the Hochschwab spring (LKAS2). Yellow circles indicate the locations of Fig. 2, 3, and 5.**

# 3 Method of process based mapping of surface runoff propensity

The proposed method of mapping runoff characteristics consists of combining (i) prior spatial information (e.g. geological map, digital elevation model), (ii) a visual assessment from a distance at the landscape scale (e.g. traces of surface runoff, spatial extent of hydrologically homogeneous areas) and (iii) local data collection in the field (e.g. soil moisture, grain size distribution). The mapping results in a classification of the entire landscape which is used for calculating a surface runoff propensity index. The mapping method was developed and tested for the Hochschwab area.

Before starting the field campaign, available spatial information of the study region was analysed. A LiDAR data set at 1m resolution was used to obtain a prior understanding of the geomorphology of the area (including karst forms), and for orientation in the field. A hydrogeological map was analysed to obtain an understanding of the bedrock (Stadler and Strobl 2006, based on geological and tectonic information by Mandl et al. 2002, Decker 2006, and Bryda et al. 2013). On the basis of both maps, strategic lookout points were identified that were suitable for a landscape scale assessment. These were mainly vegetation-free hilltops or locations on counter slopes, preferably with the line of sight orthogonal to the land surface, and that were accessible during the campaign.

During the field campaign, the field person positioned him/herself at a lookout point. On the basis of a visual assessment of the landscape, its geomorphology, apparent soil characteristics, surface runoff traces, water logging and vegetation patterns ("reading the landscape") the field person then tentatively delineated polygons on the map with approximately homogenous characteristics (Hydrological Response Units, HRUs) and specified these characteristics. Next he/she selected representative points in these polygons, still from the lookout point, to test their position and characteristics. Selection and number of points depended on the size of the polygon (more points in larger areas), line of sight from the lookout point (no interruption by vegetation) and the spatial patterns of similarity indicators such as morphology and slope. He/she then walked to these representative points and performed local measurements (soil depth, grain size, soil moisture, ...) which they used to revise the polygons or characteristics, if necessary. The procedure was repeated starting with the next lookout point. In some cases it was found more practical to apply the method in reversed order. Representative points, i.e. positions where a significant change in properties were found while walking on a slope, were mapped locally and the transfer to the area was done afterwards from a lookout point.

The mapping of the polygons, or HRUs, involves four characteristics, and each polygon consists of a combination of these characteristics (Tab. 1):

- geology
- soils and debris
- vegetation, and
- infiltration capacity.

(a) Geology (3 classes): Three geological classes occur in the study region according to the geological map: limestone (KAL), dolomite (DOL) and calcareous sand stone (KSS). The three classes mainly differ in their karstification potential. KAL is much more karstified than DOL and KSS and thus, cracks and fissures predominantly lead to immediate percolation into the karst system. DOL and KSS bedrock often show ponding characteristics which favours the generation of surface runoff and lateral subsurface flow towards a drainage network. The classification was based on the hydrogeological map and was checked during the field campaign, and was complemented by an assessment of the karstification capacity and tectonic stress related to water storage capacity based on the Quality Designation Concept. Finally, the infiltration capacity of the

bedrock was classified into high, medium and low on the basis of the geological classes and the additional hydrogeological assessment above for each polygon.

(b) Soils/debris (8 classes): The main mapping item was the type of the loose material, debris and soils along with their
infiltration characteristics. Grain sizes were visually assessed. One soil sample per site at selected mapping points was sieved to obtain information about water storage capacity and infiltration capability, using a 2 mm sieve to separate sand and gravel (i.e. fraction of material larger than 2 mm in vol.-%). The 2 mm threshold was chosen as it lends itself to dry sieving and because of its frequent use in soil mapping as a threshold between fine and coarse material (AG Boden, 1994). These measurements are quick to make and require a minimum of equipment. A total of 26 points was selected for detailed
analyses during the field campaigns. Soil depths were mapped based on visible soil profiles at slope cuttings. They were classified into four groups (<0.1m, 0.1-0.5m, 0.5-1m, >1m). Deeper soils indicate larger storage capacity and delayed runoff. Table 3 provides a summary of the soil/debris type classification.

An important item in this study was the mapping of surface runoff traces at hillslopes, such as washed out sediments and bent vegetation. Characteristics related to surface runoff were visually mapped from a distance, particularly observable
permanent flow (drainage network and springs), indications for temporary surface flow, such as dry creeks, bent vegetation (post event), erosion traces on the hillslopes or sediment that accumulated during surface flow. The observations on a particular day reflected all visible traces from the events during a period preceding the mapping date. Some of the surveys were performed immediately after heavy storms (e.g. a storm on 4 August 2013) which allowed mapping of particularly clear traces of surface runoff (post event surveys, see e.g. Borga et al., 2008). Intense and large storm events may also leave
traces of infiltration excess runoff on the highly permeable debris. Temporary and permanent creeks were visually mapped (also from a distance), and the order of magnitude of the flow width was estimated visually (0m, 0-10m, 10-100m, >100m). Water logging occurrences, also by mapping vegetation wetness indicators (Markart and Kohl, 2004; Rogger et al., 2012), assisted in identifying areas where saturation excess is deemed to occur frequently. Water logging and surface runoff traces are related to the soils and were therefore used as one of the indicators for the soil classification.
Another spatio-temporal characteristic variable is soil moisture, which was measured during the field campaigns by portable TDR ("ThetaProbe", UMS) equipment down to 6 cm depth. Standard equations recommended by the sensor producer were used to transform the measured dielectric constant to volumetric water content for organic and mineral soils (UMS, 2001). Measurements at characteristic points were performed primarily to confirm (or otherwise) the assessed drainage characteristics of the corresponding polygon. The measurements were conducted to assess the spatial variability of soil
moisture (e.g. 7 points in the Spitzboden test area) and the temporal variability (e.g. 11 points along an easily accessible transect on three dates).

Typical karst forms were identified from a distance, and the locations were cross-checked with the hydrogeological map and the 1m elevation model. Only those karst forms were mapped that were considered significant regarding surface runoff and subsequent direct infiltration. These were generally non-filled sinks with runoff traces indicating a sizeable local catchment.

Finally, the infiltration capacity of the soils/debris layer was classified into high, medium and low on the basis of the soils/debris information described above for each polygon, including general morphology (i.e. apparent karst forms), soil moisture measurements, water logging and surface runoff traces.

(c) Vegetation (2 classes): Vegetation cover was assigned one of two classes, dense vegetation (SVE) and sparse vegetation (GVE). Dense alpine forest does not occur on the plateau. Areas with dense grassland (SVE) mostly exhibit soils with significant organic matter and high moisture content. Sparse vegetation was assigned to an area not fully covered by vegetation, i.e. included areas with bare soil or bare rock. Since the campaigns were conducted in the summer and the focus was on summer conditions, the vegetation state during the campaigns was considered representative. Wetness indicators based on hydrophilic plants were mapped according to Markart and Kohl (2004) (also see Rogger et al., 2012 for application of the method). The occurrence of wetness indicators indicates low storage capacity (high local groundwater table), reduced infiltration capacity and the potential for preferential surface flow paths.

(d) Infiltration capacity (3 classes): As a synthesis of the classifications of the infiltration capacity of the bedrock and that of the soils/debris, and accounting for vegetation effects, the overall infiltration capacity of each polygon was classified in-situ by the field person (Tab. 1) into three classes: High (HIN), medium (MIN) and low (GIN) infiltration capacity. Conceptually it was considered to represent the infiltration capacity for a typical summer storm, across a polygon. This classification is an attempt of an overall evaluation of the infiltration capacity of the rock/soil/vegetation continuum in the field. The overall capacity was therefore determined as the minimum of the infiltration capacities of the two layers, soils/debris and bedrock. For example, low infiltration capacity of the soils/debris and high infiltration capacity of the bedrock was considered to result in an overall low infiltration capacity. High infiltration capacity of the soils/debris and medium infiltration capacity of the bedrock was considered to result in an overall medium infiltration capacity. The minimum was chosen as an approximation to the effective hydraulic conductivity of a layered porous medium with flow perpendicular to the layers which is the harmonic mean of the individual conductivities. To account for the effect of vegetation on the infiltration capacity, in case of dense grassland (SVE) the overall infiltration capacity was reduced from high to medium (the other two classes were not changed). This change reflects the uptake of water by the vegetation and enhanced surface runoff on mountain meadows (Markart and Kohl, 2004). In some areas, the classification was clear. For example, on the high alpine, intensely karstified plateau with no significant soils, no vegetation, but clearly visible karst forms, the high infiltration capacity classification was obvious. In other areas, the characteristics within the polygons were heterogeneous, so the classification was more difficult. Typically, units with a scale of more than 100 m were resolved (e.g. a debris area of 200 m length) while smaller units were lumped into the same polygon (e.g. an area with debris at the foot of a slope of 20 m length). This synthesis step also allowed a plausibility check of the polygon delineation in terms of the homogeneity of the polygons and their mapped properties.

This approach contrasts with the traditional geological mapping paradigm which consists of fully walking the region. The proposed approach uses variables that can be assessed from a distance in the landscape. While the accuracy may be

somewhat lower, it enables the rapid mapping of larger areas. In this study, a total area of 63 km² was mapped by one person within a total of 12 days, a two-day campaign in July 2012, and a campaign from 1 July to 6 September 2013.

**Table 1: Process oriented mapping catalogue involving geology, soils and debris, vegetation, and infiltration capacity. x indicates whether information from existing data, local mapping or from a distance has been used. * indicates that information has been combined from the previous rows.**

| Main property | Item | Mapping | | | Synthesis | Method | Examples | Processes |
|---|---|---|---|---|---|---|---|---|
| | | existing data | from a distance | local mapping | | | | |
| Geology | Classification | x | | x | | Geological map, randomly checked in the field (hydrochloric acid test) | Limestone (KAL), Dolomite (DOL), Calcareous sandstone (KSS), 3 classes | Percolation into karst system vs. lateral subsurface flow towards drainage network |
| | Assessment | x | | x | | Rock Quality Designation | Karstification capacity, tectonic stress related to water storage capacity | Percolation into karst system vs. lateral subsurface flow towards drainage network |
| | Classification of infiltration capacity of the bedrock | | | | * | Synthesis of geological information | 3 classes: high - medium – low infiltration capacity of the bedrock | |
| Soils and debris (surface material) | Grain sizes | | | x | | Descriptive | Silt, sand, gravel, etc. | General soil physical properties |
| | Fraction larger than 2 mm | | | x | | In-situ sieving at selected mapping points | | Reduced storage capacity and increased hydraulic conductivities |
| | Soil depths | | | x | | Morphological estimates based on visible soil profiles at slope cuttings | Deep, shallow | Increased storage capacity, runoff delay |
| | Surface runoff traces | | x | x | | Visual examination | Gullies, erosion traces, sediments, bent vegetation | Surface runoff generation propensity and frequency: temporary and permanent creeks |
| | Indication for waterlogging | | x | x | | Visual examination | Swamps, ponds | No/very low storage capacity, immediate surface runoff generation |
| | Soil moisture at time of survey | | | x | | TDR (portable) | Relative differences between mapping points (spatially) and between different dates (temporally) | Distinction between generally wet or dry areas (spatially), wetting and drying dynamics (temporally) |
| | Karst forms and surface runoff towards these | x | x | x | | Visual examination, assisted by 1m DTM and hydrogeological map | Dolines, sinks | |
| | Classification of soil type | x | x | x | * | Based on soil and debris descriptors | OLG, BOF, HSG, etc. (8 classes), see Tab. 3 | |

| | | | | | | | | |
|---|---|---|---|---|---|---|---|---|
| | Classification of infiltration capacity of the soils/debris | | | | | * | Synthesis of soil and debris descriptors | 3 classes: high - medium – low infiltration capacity of soils/debris | |
| Vegetation | Classification | | x | x | | | Visual examination | 2 classes: GVE, SVE | Interception and ET processes (of minor relevance in this work) |
| | Wetness indicators | | x | x | | | According to Markart and Kohl (2004) | Hydrophilic plants | water logging, high water availability for plants (e.g. high local groundwater table), low storage capacity, preferential surface flow paths |
| Infiltration capacity | Classification | | | | | * | Combination of infiltration capacity classes of the bedrock with those of the soils/debris, accounting for vegetation effects | 3 classes: high - medium - low overall infiltration capacity (HIN-MIN-GIN) | |

# 4 Examples of mapping results

Figure 2 shows an example of the mapped polygons. Polygons no. 1, no. 4 and no. 7 consist of limestone (KAL) and differ in their overlying material/debris: in polygon no. 1 bare rock occurs, no. 4 is a rockfall area and no. 7 is flatter and exhibits a

thin soil layer. Soil samples indicated fine soils (BOF, Tab. 3). All three polygons show sparse vegetation (GVE). No. 13, no. 16 and no. 17 are examples for underlying calcareous sandstone (KSS) with low permeabilities compared to KAL. They differ in their slopes which has led to different soil sedimentation processes. The flat area in no. 17 shows organic soils with high wetness (BOO). A drainage network originates in this polygon, which forms a permanent creek. In polygon no. 13 much less organic material has been deposited, but there is a high fraction of fine material in the soil (BOF) (identified by

detailed mapping). Polygon no. 16 is a rock fall area, but the vegetation cover (SVE) indicates some water storage in the debris fed by runoff from the adjacent bare rock areas and supported by the ponding properties of the underlying KSS. Except for no. 17, surface runoff traces such as gullies are not pronounced in the polygons, which indicates low surface runoff propensity and significant deep percolation.

Figure 3 shows a cross section of the mapping results in the Spitzboden area (see Fig. 1). Bare dolomitic rock at the

Plankogel (far SW, left in Fig. 3) shows low infiltration capacity, because of its small fissures. Precipitation falling on these areas will mostly flow towards the debris and the drainage network on the surface, due to the steep slopes. Only at high precipitation intensities will water flow through the debris and reach the drainage network as surface runoff, otherwise it is expected to infiltrate into the debris and contribute to the springs. In these ditches connected to the neighbouring rock walls only occasional surface runoff traces were found in the field. The springs form a drainage system and a small creek, which

sinks into a karst doline right after the stream gauge location. The bottom of the Spitzboden (centre in Fig. 3) is dominated by low permeable sediments overlain by soils with high organic contents. A permanent drainage system exists, which

facilitates surface and subsurface flow generated by the precipitation on the area itself and subsurface flow from the neighbouring dolomitic debris. Due to the available but low storage capacity of the organic soil (BOO), surface runoff is expected to be slightly delayed (saturation overland flow). On the neighbouring debris, temporary surface runoff traces (small ditches) were observed, but springs at the bottom of the slope indicate that, most of the time, precipitation infiltrates

5 and subsurface flow tends to occur at the interface to the relatively low permeability underlying dolomite (interflow, shallow groundwater flow). Permanent spring discharge could be observed which contributes to the Spitzboden drainage network. However, the small ditches indicate surface runoff towards the lower permeable organic soils, i.e. the network, at high rainfall intensities. Rocks in the northeastern parts (right in Fig. 3) are calcareous sandstones and limestones with high infiltration capacities. From the Spitzboden to the foot of the Schaufelwand the bare rock is covered by coarse debris (gravel)

10 and rockfall material with very high permeability. It is very likely that precipitation falling on these areas contribute very little to the Spitzboden creek. Direct infiltration is dominant with subsequent deep percolation into the karst system. Exceptions are the very steep slopes where surface runoff can occur but will infiltrate immediately into the debris at the slope bottom, as inferred by the short flow lengths of the gullies observed.

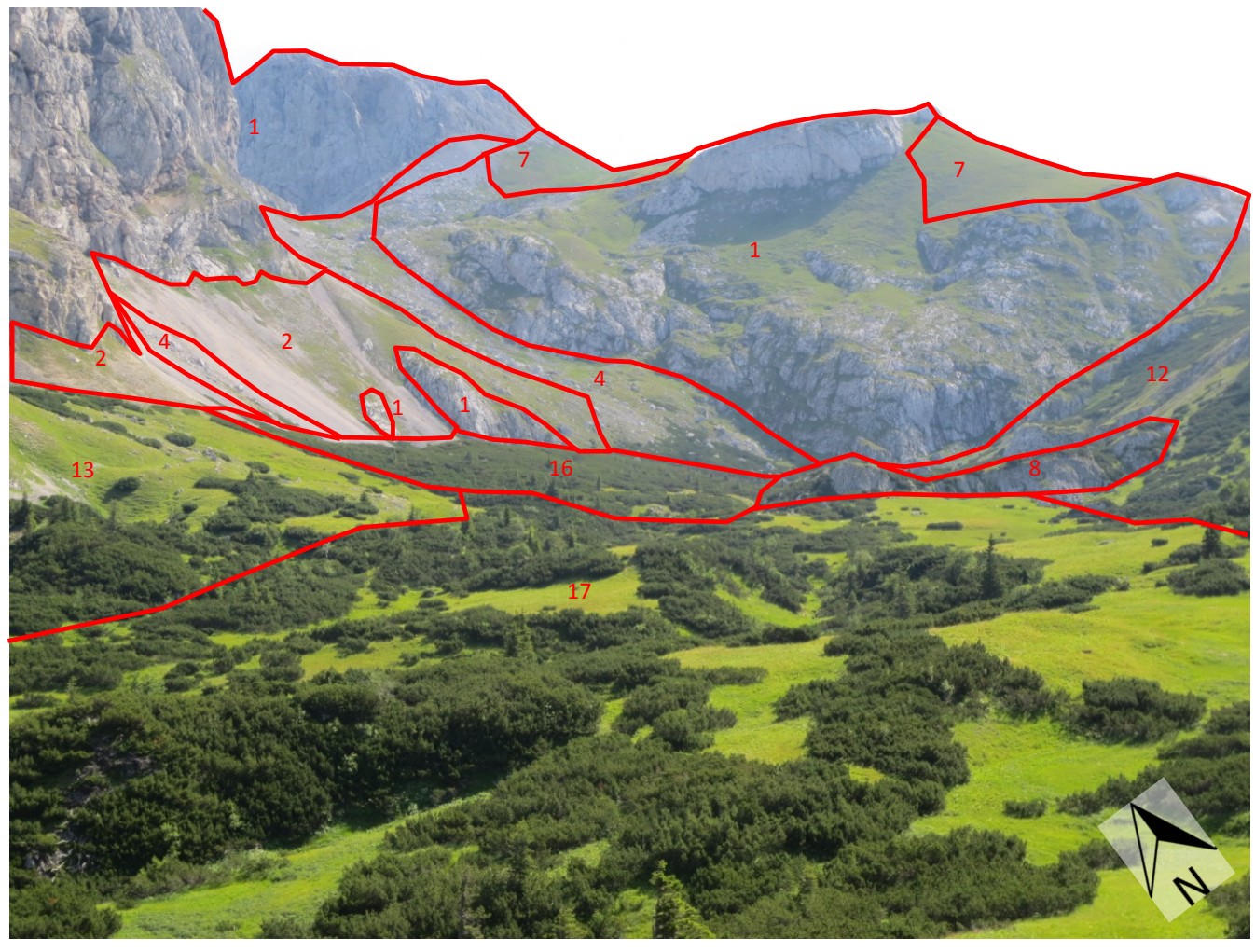

**Figure 2: Example of delineation of response units based on Tab. 1. Explanation of units see Tab. 2. For illustrating the scale: The diameter of polygon no. 16 from left to right is app. 300m.**

5    **Table 2: Hydrological Response Units of Fig.2.**

| No | Geology | Soils, debris | Vegetation | Infiltration capacity |
|----|---------|---------------|------------|----------------------|
| 1 | Limestone | Bare bedrock | Sparse | High |
| 2 | Limestone | Coarse debris | Sparse | High |
| 4 | Limestone | Boulders | Sparse | High |
| 7 | Limestone | Loam | Dense | Medium |
| 8 | Dolomite | Bare bedrock | Sparse | Low |
| 12 | Dolomite | Loam | Dense | Medium |
| 13 | Calcareous sandstone | Loam | Dense | Medium |
| 16 | Calcareous sandstone | Boulders | Dense | High |
| 17 | Calcareous sandstone | Organic soils | Dense | Low |

**Table 3: Classification of "soils/debris" type into 8 classes.**

| Type (Acronym) | Name | Description |
|---|---|---|
| OLG | Bare rock | Bare rock (no soils), properties of geology control runoff generation |
| HSF | Fine debris | Debris predominantly fine material from dolomitic rocks, low permeability, medium to high storage capacity |
| HSG | Coarse debris | Debris predominantly coarse material, often at the bottom of steep calcareous slopes |
| BST | Boulders | Rockfall material: large boulders, infiltration properties of underlying geology (mainly limestone) becomes dominant |
| FLS | Fluviatile sand | Fine fluviatile sediments (small depth), connected to permanent water flow (creek bed, riparian zone), occurs rarely |
| BOF | Loam | Soil with high fraction of fine material, mainly Neogene ("red loam", secondary sediments from crystalline), comprise organic material, potential humus, grain size: silt/clay, very low permeability |
| BOG | Coarse sand | Soil with low/no fraction of fine material, in-situ developed mainly over limestone, small depths, occurs also at steeper slopes, very low storage |
| BOO | Organic soil | Soil with significant organic components (humus), mainly above BOF with shallow impermeable layers, low storage |

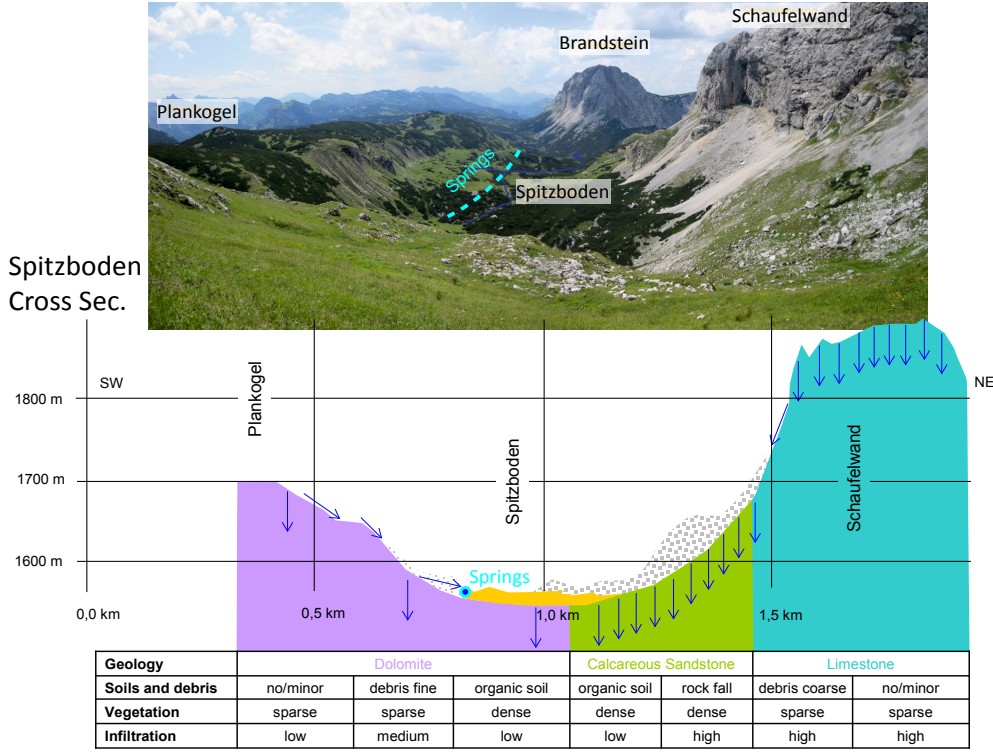

**Figure 3: Cross section of the Spitzboden area illustrating the mapped properties (geology, soils and debris, vegetation and infiltration capacity, see Tab. 1). Yellow and grey shading relates to organic soil layer and coarse debris from the wall in the Northeast (NE), respectively. Blue arrows indicate dominant drainage directions during a typical summer storm; a higher density of arrows indicates larger fluxes.**

# 5 Definition of the surface runoff propensity index

An index is proposed based on the mapped infiltration capacity to summarize typical patterns of surface runoff. We term this index "surface runoff propensity index". It is intended to reflect the likelihood of a particular location to exhibit surface runoff during high intensity summer storms. Since the rainfall mechanisms are rather uniform in the study area, the index focuses on the infiltration capacity of the soil as well as topographic slope. The latter reflects the field observation in the study area that steep slopes tend to exhibit surface runoff more frequently than flatter slopes with the same soil types. Even in intensely karstified areas with coarse debris on steep slopes surface runoff was observed to occur. Traces (gullies) were observed resulting from runon from bare rock areas above and surface runoff generated on the area itself. For example, Fig. 5 shows traces of a temporary flow path above highly permeable sediment after the extreme event on 4 August 2013. The flow path collected surface runoff from the steep upslope areas (bare rock and debris) associated with the infiltration excess mechanism. Incorporating topographic slope in the surface runoff propensity index is mainly intended to represent the steep slopes of karstified bare rock and debris with high and medium infiltration capacities. For areas with low infiltration capacity, slope was not used in the index.

The index values are estimated by a decision tree from the classified infiltration capacity and topography (Fig. 4). Four classes were defined: no surface runoff, low, medium and high surface propensity. No surface runoff can be expected at locations with coarse sediments/debris and intensely karstified rocks with high infiltration capacity (HIN) in relatively flat areas. The surface roughness of these areas is high, and rainfall as well as rare infiltration excess surface flow will infiltrate immediately. The field mapping and morphologic analyses showed that 20° is the range of threshold slopes beyond which gullies occur much more frequently, so the threshold slope was set to 20°. Low propensity is assigned to areas with intensely karstified rocks with high infiltration (HIN) and steep slopes (slope > 20°) as well as areas with medium infiltration (MIN) and flat slopes. Only short gullies can sometimes be observed on steep slopes in these areas. Clear surface runoff traces were not observed, but ephemeral surface runoff with very short flow lengths (<10 m) and subsequent infiltration in adjacent areas may sometimes occur. Medium propensity is assigned to areas with medium infiltration (MIN) and steep slopes. Traces of temporary surface runoff are clearly evident (gullies with surface flow lengths of up to about 100 m) in these areas. High propensity is assigned to areas with low infiltration (GIN), showing a pronounced drainage network with surface flow lengths generally much larger than 100 m. This class is independent of the slope.

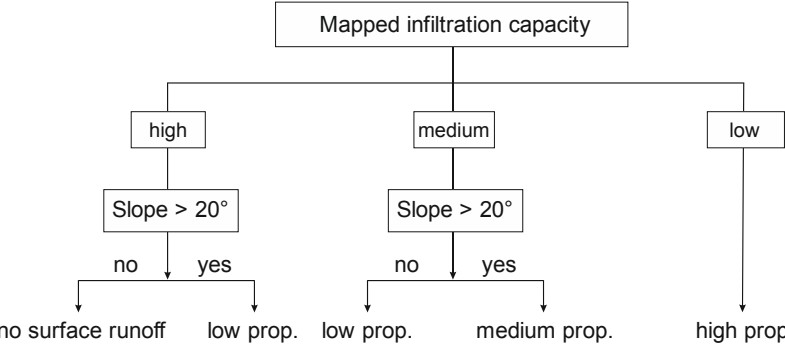

**Figure 4: Decision tree for assigning the propensity index to the mapped polygons of infiltration capacity.**

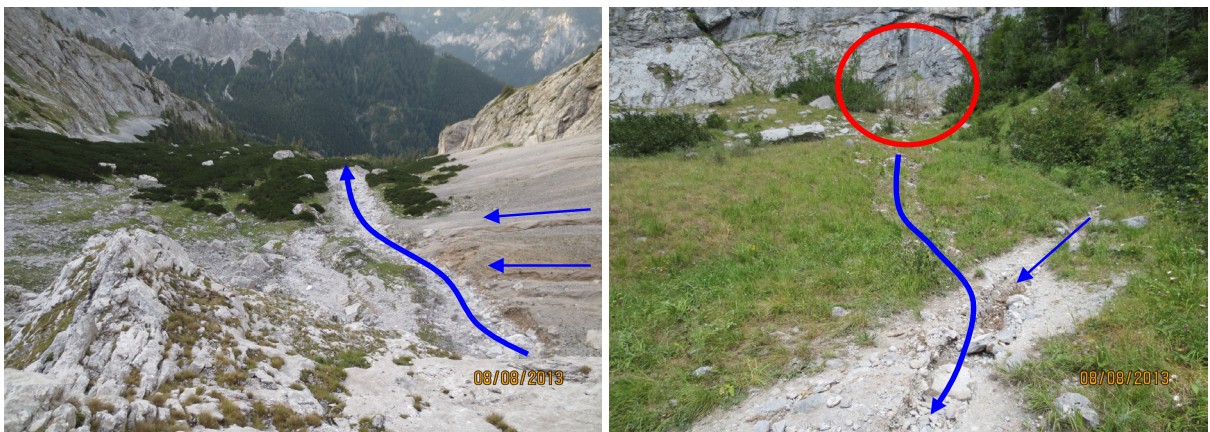

 **Figure 5: Examples of traces of surface runoff paths from the extreme event on 4 August 2013 (photos taken 4 days after event at location indicated in Fig. 1). Thin blue arrows: surface runoff traces on highly permeable gravel/debris; thick blue arrows: larger surface runoff paths, collecting runoff from larger areas during the event. The main water flow path in the left photo ends at the edge of the rock wall, and the right photo shows the situation at the foot of the wall. The red circle indicates the leafless vegetation caused by a temporary waterfall (water and debris). For illustrating the scale: The flow length of the main flow path in the left**
 **photo is app. 20m.**

# 6 Testing the surface runoff propensity index

The spatial pattern of the surface runoff propensity index (4 classes, Fig. 4) was tested by comparing it with patterns of sink holes identified from the digital terrain model (DTM). The sink holes were identified by aggregating the 1m DTM to 3m and delineating pixels without outflow. The sink hole distribution was not used for estimating the index, so can be used for testing it. The index was processed on a raster with a cell size of 10 m.

Figure 6 shows a detail of the study area including the Spitzboden catchment where permanent runoff was observed. The location of the sinks (plot b) corresponds very well with the areas of index no. 1, which indicates no surface runoff. High propensity is assigned to the areas with less karstified bare rock (DOL), sometimes overlaid by low permeability debris and

organic soils. In these areas surface runoff can occur very often and, depending on the morphology, a permanent drainage network is likely to develop. Even formation of ponds and small lakes, such as the one in the southeastern part of the test catchment, is possible. In the large karstified areas the distinction between low (no. 2) and medium (no. 3) is mainly a result of differences in the topographic slope (Fig. 4). In these areas only temporary surface runoff occurs with very short flow lengths until the flow sinks into highly permeable debris or dolines.

In Fig. 7 the surface runoff propensity index in the whole study area (Hochschwab region, plot a) is plotted and compared to the sink holes (plot b). No surface runoff is calculated in 11 % (7 km²), low propensity in 65 % (41 km²), medium propensity in 21 % (13 km²), and high propensity in 3 % (2 km²) of the study area.

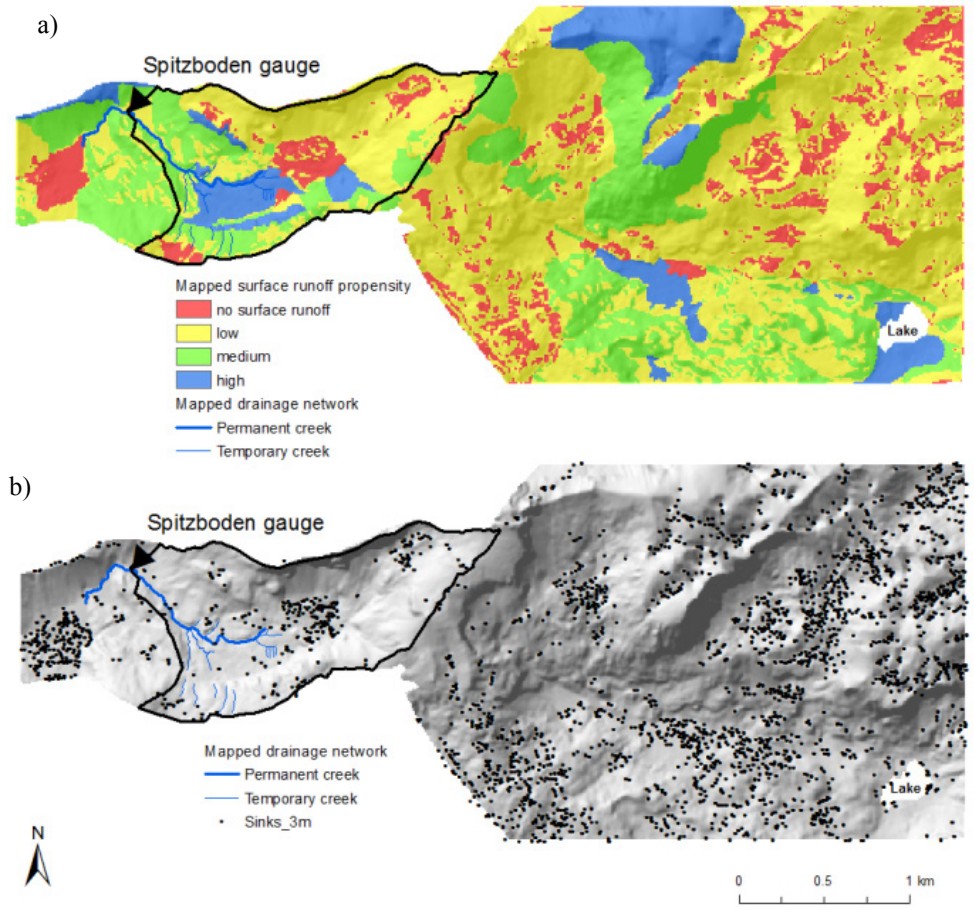

Figure 6: Comparison of the surface runoff propensity index (a) with sink holes (b). Sink holes have been automatically generated from the LiDAR terrain model at 3m resolution. Shading represents topography from the 1m terrain model. Black polygon shows the topographic catchment of the Spitzboden stream gauge.

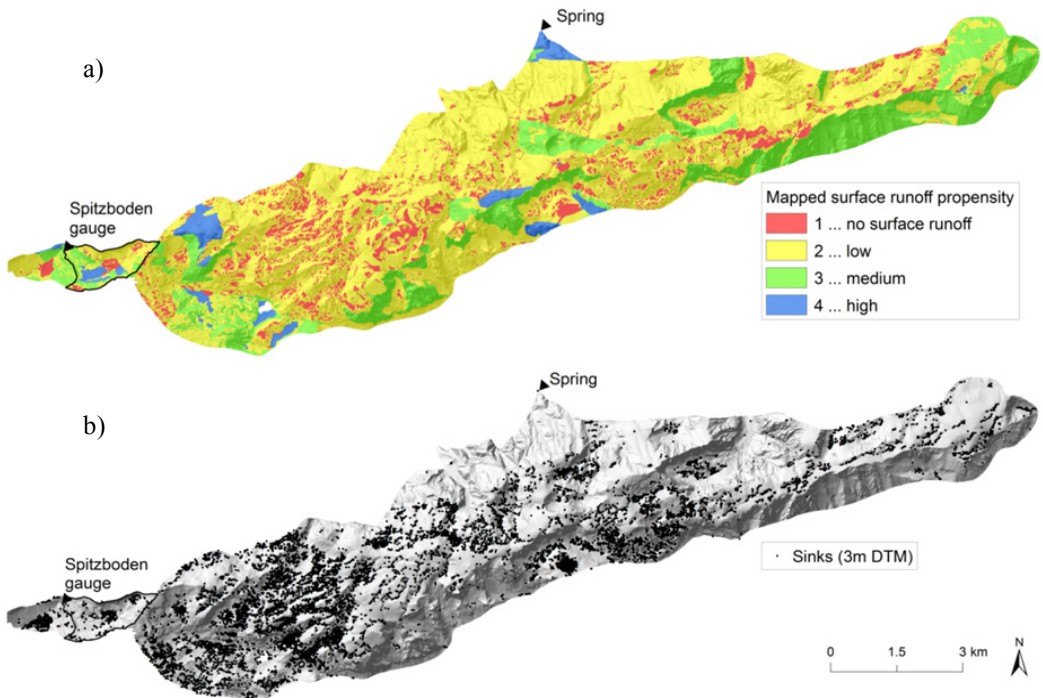

**Figure 7: Top: Surface runoff propensity index in the whole study region (Hochschwab region, a). Bottom (b): sink holes from the terrain model. Shading represents topography from the 1m terrain model.**

# 7 Discussion

## 7.1 Strengths and limitations of the method

The main advantage of the proposed mapping method over traditional field mapping (e.g., Scherrer and Naef, 2003) is its ability to use variables that can be assessed from a distance in the landscape, so large areas can be mapped efficiently. The main advantage over traditional vulnerability mapping (e.g., Cost Action 620, 2003; Andreo et al., 2006) is the incorporation of hydrological processes in the mapping.

Rogger et al. (2012) used geologic maps, orthophotos and geomorphic information to identify the dominant subsurface processes in Alpine catchments. These were combined with the surface runoff process maps of Markart et al. (2004) that are based on sprinkling experiments using vegetation and soil characteristics as indicators for event runoff coefficients. Sprinkling experiments are logistically challenging in remote alpine areas of the size of the study area. Instead, the mapping method in this study puts more emphasis on "reading the landscape" for delineating Hydrological Response Units (HRUs) and classifying them according to geology, soils/debris, vegetation and infiltration capacity. Spotting karst forms and surface runoff traces are an important part of the method. The field mapping is supported by morphological and geological maps, but

they are not directly used for HRU delineation as in traditional vulnerability mapping. Additionally, the location of springs provides in situ information on the hydrogeology (see, e.g., Cervi et al., 2017).

The mapping catalogue proposed here is defined in a way that it maximizes the information that can be mapped in the field with a view of surface runoff propensity, including infiltration/storage characteristics, immediate response vs. delayed response. Of course, like in any field mapping method, some subjectivity is involved but minimized by the use of a catalogue. Used as a "recipe" it will provide similar but not identical polygons if used by different field persons, due to differences in the path choices of the individuals and selection of representative points to verify the properties. Also, the mapping scale has an element of subjectivity, i.e. which sub-scale variability is allowed. However, a careful application of the method should provide very similar areas with high propensity (e.g., bare dolomitic rock with low infiltration capacity of the small fissures and permanently wet areas). The date of the field work may sometimes influence the results as visual traces of surface runoff are related to intense rainfall events in the preceding weeks. We would like to point out that alternative methods usually also involve a certain level of subjectivity in terms of choice of variables and model structure. In vegetation mapping, for example, due to the typically large diversity it is difficult to assure that all species within an area are resolved. A trade-off always exists between mapping effort and mapping scale (resolution), particularly when mapping large areas, and every method has limitations in terms of accessibility and visibility in remote regions.

We used the mapping results for estimating a surface runoff propensity index at the pixel scale (10 m x 10 m) that is a measure of the frequency with which surface runoff occurs. Although the pixel scale is 10 m, the aggregation scale is larger as it is mainly controlled by the size of the mapped polygons which are on average 32 ha in area. The hydrological characteristics do vary within the polygons. This means that the smaller scale spatial variability is not explicitly represented by the index.

The index also points towards possible flow lengths as typically, areas of more frequent runoff generation in the study region are also those with larger flow lengths. The propensity index is derived from the mapped infiltration capacity and topographic slope. It compares well against the sink hole distribution. The index can be used for a rapid regional assessment of contamination risk for water resource purposes. Additionally, the index could be used for supporting the parameter estimation of a rainfall-runoff model in the context of the "Dominant Processes Concept" (e.g., Grayson and Blöschl, 2000; Reszler et al., 2008) and more generally for regionalising hydrological characteristics related to surface runoff (Blöschl et al., 1995; Blöschl, 2006).

## 7.2 Surface runoff generation processes

In karstic areas, surface runoff occurrence depends on the infiltration properties of the soil, debris and the underlying geology. In the study area, organic soils have mostly developed above poorly karstified rocks and, hence low permeability lithology such as dolomite. In these areas, a permanent drainage network is prone to occur, the soils have small storage capacities and water logging, including the formation of ponds, is likely to occur, leading to frequent surface runoff. Both the

local drainage network and the ponds drain into the karst. Processes on these areas have substantial similarity to the Saturation Overland Flow mechanism (Dunne, 1983), which is common in humid climates at the foot of hillslopes as they flatten out. Such areas are important to identify, but their extent in karstic catchments are sometimes small. In the Hochschwab study region, 3% of the area was classified as high surface runoff propensity based on the index. When substantial debris is found above the ponding layer, mostly in dolomitic areas with higher slopes, local springs indicate the dominance of storage and subsurface flow that exfiltrates and feeds the local drainage network. Traces of surface runoff paths above the debris were visible during the surveys, but pronounced continuous flow paths were hardly found. This implies that flow connectivity to the creek is only established at high rainfall intensities.

The mean altitude of the recharge area of the spring is about 1700 m a.s.l. and soils are shallow and sparsely vegetated. In some parts of the area, losses due to evapotranspiration may play a role for the general soil moisture status in summer (e.g. in areas with high local groundwater levels, water-logged areas). These characteristics are captured by the mapping and are classified as a relative high surface runoff propensity. It is possible that, after long dry periods, these areas are drier than predicted by the index. However, given the depth to the groundwater table in most of the area, long-term spring water balances could not be used to validate the index (Bonacci 2001b).

In the highly karstified areas of the study region, often overlaid by debris and/or a thin humus layer, surface runoff occurs only during high intensity storms and is therefore very rare. This is consistent with the literature (e.g., Zhang et al. 2011; Peng and Wang 2012). The storm of 4 August 2013 (return period of about 10 years) allowed the mapping of particularly clear surface runoff traces during a post-event survey. Rainfall intensities of about 100 mm in 75 minutes were recorded at the nearby stations, which is on the order of the rainfall necessary for generating surface runoff obtained by sprinkling experiments in semi-arid karstic environments (Li et al., 2011; Calvo-Cases et al. 2003). The index results suggest that in about 50% of the area no surface runoff occurs or the surface runoff propensity is low. However, some high intensity (local) rainfall events are likely to occur every summer, which do produce surface runoff even in areas that are highly karstified.

## 7.3 Transferability of the method

The index proposed in this paper could also be adopted for non-karstic regions. The overall methodology of combining prior spatial information, a visual assessment from a distance at the landscape scale and local data collection in the field, as well as the combination of geological, soils/debris and vegetation information may remain similar. However, the individual geological, soils and vegetation classes will obviously change. Also, some of the field instrumental methods may change, depending on the local hydrology, as other processes may gain more importance, e.g., lateral subsurface flow, shallow groundwater level fluctuations, connectivity of flow paths, runoff routing. Of course, stream gauging would become more important in non-karstic areas. Runoff dynamics may point towards possible runoff components, and water balance considerations relying on accurate catchment boundaries may indicate losses by evapotranspiration that can assist in

estimating soil moisture dynamics during the period of field work. Also, surface runoff traces will be much more frequent in non-karstic areas, and capturing events by post-event surveys will become more likely.

In any case, there is a very important role of the field persons in assessing the hydrological situation. Even though this implies some subjectivity, we believe the additional information obtained during the field campaigns outweighs any
subjectivity. Aerial photos such as from drones could be useful to resolve more spatial detail than the aerial photos used here, but visibility in the forests is an issue.

# 8 Conclusions

A new method for mapping the propensity for surface runoff in karst areas is proposed. The method puts emphasis on "reading the landscape". It includes information that can be assessed from a distance, such as visible traces of overland flow,
so large areas can be mapped efficiently. The outcome of the mapping is a tessellation of the landscape into Hydrological Response Units, based on a classification of the geology, soils and debris, vegetation, and infiltration capacity. The method was applied to the karstic Hochschwab area in Austria. In this region, surface runoff is generally more frequent in less karstified dolomitic areas than in limestone areas. Low permeability soils or debris are more likely to occur above dolomitic geology, where a permanent drainage network tends to develop. A "surface runoff propensity index" is estimated from the
classified infiltration capacity obtained by the mapping. It represents an index of how frequently surface runoff may occur. The index compares well with the spatial distribution of sink holes. The surface propensity index can be used to assist in safeguarding the quality of the water supply from karst aquifers, particularly for optimizing land management and formulating water safety plans in a risk based procedure by comparing the patterns with potential contamination loads. Since the overall conceptualization of the mapping procedure is generic to karst areas, it may be applied to other karst regions
around the world.

**Acknowledgements**

This paper is dedicated to the memory of our wonderful colleague, Dr. Hermann Stadler, who recently passed away. Funding from the Vienna Water Works through two projects of "Mapping and modelling spatial patterns of surface runoff in the
karstic catchments Hochschwab and Zeller Staritzen" is gratefully acknowledged. We would like to thank Gerhard Kuschnig and Wolfgang Zerobin for their support of these projects. Funding from the Austrian Science Foundation as part of the Vienna Doctoral Programme on Water Resource Systems (DK Plus W1219-N22) is also acknowledged.

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
