# Peer review of "A propensity index for surface runoff on a karst plateau"

_Hydrology and Earth System Sciences, 2018_

## Referee Comment (RC1) · Anonymous Referee #1 · 28 Apr 2018

Relationship between the propensity index for surface runoff and effective infiltration coefficient (runoff coefficient) should be better explained or at least more detailed discussed. The consultation of the paper "Monthly and annual effective infiltration coefficients in Dinaric karst: example of the Gradole karst spring catchment" published in Hydrological Sciences Journal 46:2, 287-299, DOI: 10.1080/02626660109492822 will be useful for this. Role of vegetation in different seasons should be better explained. In karst terrains very important role plays groundwater level. This fact has to be stressed.

---

## Referee Comment (RC2) · J. Juilleret (Referee) · 30 Apr 2018

I enjoyed to read this paper concerning the development of an index for assessing runoff in karst areas. The index integrate a fundamental point in geoscience consisting of "reading the landscape". In this paper, the authors, made an effort to compile and propose what they have in mind (even if it looks sometime subjective) when they build their perceptual hydrological model. The authors developed their index by including geology, pedology, hydrology that consist of field approach and mapping to delineate their HRU.

1) Introduction Despite the effort, I found that the paper lack the "state of the art"on HRU, consequently I suggest that the authors highlight the previous similar studies

concerning HRU and how different sources of information can contribute to the development of perceptual models in their introduction. I kindly ask them to better explain difference between HRU and HSA, limits and advantage of the HRU approach. I would like that the authors better explain why their method is innovative and better adapted in karstic and/or mountainous environment. Link to that, explain why traditional methods tend to focus on geology, geomorphology and not soil. What is the reason? (Suggestion: lack of soil map, if maps exist difficulties to translate the soil map unit classification into pedohydrological concept). Please provide references in your general assertion like karst water is used for water supply in Vienna, in karstic areas soils tend to be shallow, etc...

2) Case study area Please specify where the Hochschwab massif is in a general map representing Austria. Add references on the vegetation cover. I miss a more information (map) from the key lookout points in the polygons (number, position). I don't consider this request as mandatory, however it will be appreciate, indeed if the information is available, it will help to appreciate the advantage in time and space to cover the entire zone.

3) Method of process based mapping of surface runoff propensity Inverse the order (i), (ii) and (iii). Indeed, in the article you follow the inverse order.

a) Geology I suggest that the authors develop the methodology to assess the karstification potential of the bedrock. I understood that the "Rock Quality Designation (RQD)" was used but the link between this method and the 3 geology classes (limestone, dolomite, calcareous sandstone), however it is unclear. Provide more details.

b) Soils. Please better explained the "visual assessment" of the soils. What is the threshold between deep vs shallow soil? Provide a thickness range. Do they authors assess the texture of the soil (clay, silt, sand)? Or do they just classify between fine (< 2mm) and coarse (> 2mm)? If so; why do they consider that it's important? Why the authors did not used the available soil map? Why the author did not use of soil survey

to complement the visible soil profiles in the polygons? I think that it would have been be more relevant than the use of the TDR Please explain better how you obtain the three infiltration capacity classes, it's not clear.

c) Vegetation. Please provide more information on your choice between dense and sparse. Any references?

d) Infiltration capacity. I agree that there is always subjectivity when the field experimentalist propose his perceptual model of a catchment, but in your case you should make effort in such a way that everybody can reproduce your decision tree. This paragraph lack the methodology on how the authors make the synthesis of the mapped properties. A clear methodology should be provide to obtain the different classes of each polygon.

Figure 2 and 5: add scale and orientation of the view

Figure 3 : please make effort to have a real geological crossection, bedding and fault are missing and the vertical limits between Dolomite, Calcareous Sanstone and Limestone units seems not natural to me.

7) Discussion.

Provide reference on the assertion that surface runoff propensity is at the pixel scale (10 m* 10 m). Discuss not only the advantage of the methodology but also the limits (like subjectivity, compilation of knowledge in different disciplines) and how can you overcome the limits. You should also go beyond the case of study, explaining how field experimentalist are still essential into the understanding of catchment behavior, showing that even if there is subjectivity in their assessment, the role of the field experimentalist is crucial to the hydrological community in the understanding of hydrological basin and their prioposition of perceptual models.

8) Conclusion.

Please provide some widening ideas on your method. As example, in flat areas the

field person can be help by the use of drones, consequently the method can also be apply in other non-mountainous context. Provide some advice on how the method can be adapt in non-karstic context.
* * *

---

## Referee Comment (RC3) · Anonymous Referee #3 · 6 Jul 2018

General comments: The authors propose a new mapping approach for karst environments, based on local data in combination with visual mapping. The method offers potential for a broader use as all mapping variables can be assessed from distance and derived easily from DTM and geological maps. However, some important corrections of this manuscript have to be made. Beside specific comments listed below, a major aspect is as follows: From my perspective, the present paper is mainly based on describing the methodology of this approach. A "Technical Note" format might proof more useful. The discussion section could strongly improve from discussing limitations of the proposed method more in detail. It could also address whether this method is transferable to other geological settings and catchment types.

Specific comment: Page 1, Line 10: Instead of frequency, better use "spatial distribu-

[Figure]

tion" or "occurrence". Page 1, Line 17: please put "in the study area" at the end or beginning of this sentence. Page 3, Line 15: How did you delineate the catchment extent? I assume that this is derived from GIS delineation based on the surface topography (not accounting for the subsurface catchment)? Page 4, Line 16: How was the lookout point chosen? Were there logistical constrains? Page 5, Line 1 -5: Please rephrase here. Page 5, Line 19: "show" instead of "shows" Page 8, Line 11-12: Please rephrase. Page 11, Line 8: "runoff" instead of "runon".

Figure 1: Please put the label "LKAS2" in the map and show the state Austria in a smaller map with the location of this study area. Figure 2: Why does Zone no 7 not go further downslope into Zone no 1? As the visible bedrock outcrop emerges below. Figure 3: Please enlarge labels in the cross-section Figure 6/7: Please use a) and b) labels for both subplots. Also use a colour for the Spitzboden catchment extent in the lower subplot.

---

## Author Comment (AC1) · 11 Aug 2018

Author's response (AR) to reviewer #1:

We would like to thank the reviewer for his/her insightful comments.

*RC: Relationship between the propensity index for surface runoff and effective infiltration coefficient (runoff coefficient) should be better explained or at least more detailed discussed. The consultation of the paper "Monthly and annual effective infiltration coefficients in Dinaric karst: example of the Gradole karst spring catchment" published in Hydrological Sciences Journal 46:2, 287-299, DOI: 10.1080/02626660109492822 will be useful for this.*

AR: Mean altitude of the recharge area of the spring is app. 1700 m asl and soils are shallow and sparsely vegetated. In some parts losses due to evapotranspiration (ET) may play a role for the general soil moisture status in summer (e.g. areas with high local groundwater levels, water-logged areas). These properties are incorporated into the mapping and are classified to a relative high surface runoff propensity. It can be that, after long dry periods, these areas are drier than the assumption for the index (mean soil moisture status). We will address this point and include the suggested paper into the discussion, but long-term spring water balance aspects are not an issue in the paper.

*Role of vegetation in different seasons should be better explained.*

AR: Since mapping concentrates on the summer months and heavy rainfall events, this plays a minor role for surface runoff generation. Influence of ET losses in some areas see AR above. This is an interesting point though, and we will make a comment to this effect in the revised paper.

*RC: In karst terrains very important role plays groundwater level. This fact has to be stressed.*

AR: While the karst groundwater level is very relevant for spring discharges it is not really relevant in this area for runoff generation. The karst massif reaches altitudes of more than 2200 m asl and karstification is very deep. There are a few exceptions where the local groundwater level approaches the surface, where it is accounted for by the index as described in the mapping (e.g., local springs, GW-exfiltration, water logging). Again, we will make a comment to this effect in the revised paper.

---

## Author Comment (AC2) · 11 Aug 2018

Author's response (AR) to reviewer #2 (J. Juilleret):

We would like to thank the reviewer for his insightful comments.

*1) Introduction Despite the effort, I found that the paper lack the "state of the art" on HRU, consequently I suggest that the authors highlight the previous similar studies concerning HRU and how different sources of information can contribute to the development of perceptual models in their introduction. I kindly ask them to better explain difference between HRU and HSA, limits and advantage of the HRU approach. I would like that the authors better explain why their method is innovative and better adapted in karstic and/or mountainous environment. Link to that, explain why traditional methods tend to focus on geology, geomorphology and not soil. What is the reason? (Suggestion: lack of soil map, if maps exist difficulties to translate the soil map unit classification into pedohydrological concept). Please provide references in your general assertion like karst water is used for water supply in Vienna, in karstic areas soils tend to be shallow, etc...*

> AR: We will revise the introduction to address the remarks above. The basic studies of mapping HRUs in the context of the Dominant Processes Concept (DPC) have already been referred to, and we will extend the description. We will add some more literature for a more comprehensive state-of-the-art overview. The HRU (surface runoff and contaminant transport) and HSA concepts are similar, but we have no hard data about distributed erosion paths and no hard data about contamination loads, which we will mention. A statement about the challenges with soil data and the difficulties of translating existing soil maps into dominant processes descriptions will also be added. Reference to the literature regarding the Vienna Water supply will be given.

*2) Case study area Please specify where the Hochschwab massif is in a general map representing Austria. Add references on the vegetation cover. I miss a more information (map) from the key lookout points in the polygons (number, position). I don't consider this request as mandatory, however it will be appreciate, indeed if the information is available, it will help to appreciate the advantage in time and space to cover the entire zone.*

> AR: The catchment can be identified by the coordinates in the left upper corner, but we will add a small inset map of Austria within Fig. 1 for clarity. While a number of vegetation data sets are available in Austria, their resolution is relatively low. In this study, vegetation was therefore mapped as part of the field efforts. We will add a comment for clarification. The location of the lookout points had not been logged during the survey, but that's a very good point for future applications of the method.

*3) Method of process based mapping of surface runoff propensity Inverse the order (i), (ii) and (iii). Indeed, in the article you follow the inverse order.*

> AR: The presentation will be reversed as suggested.

*3a) Geology I suggest that the authors develop the methodology to assess the karstification potential of the bedrock. I understood that the "Rock Quality Designation (RQD)" was used but the link between this method and the 3 geology classes (limestone, dolomite, calcareous sandstone), however it is unclear. Provide more details.*

> AR: Regarding karstification potential and hence, water infiltration/retention potential, three classes were identified from a regionalization procedure (RQD) out of a number of geologic

formations. This had been done in previous studies providing the "hydrogeological map". Uncertain boundaries were checked during the field mapping. A more detailed description will be added to the manuscript.

*3b) Soils. Please better explain the "visual assessment" of the soils. What is the threshold between deep vs shallow soil? Provide a thickness range. Do they authors assess the texture of the soil (clay, silt, sand)? Or do they just classify between fine (< 2mm) and coarse (> 2mm)? If so; why do they consider that it's important? Why the authors did not use the available soil map? Why the author did not use of soil survey to complement the visible soil profiles in the polygons? I think that it would have been be more relevant than the use of the TDR. Please explain better how you obtain the three infiltration capacity classes, it's not clear.*

AR: As mentioned in the manuscript, mapping of soil depths is limited to available slope cuttings (point data) and extrapolated into space by sedimentation considerations based on morphology and geology (weathering, land and rockslide). A detailed mapping at the catchment scale would be very difficult, among other things, due to floating rocks. The threshold between deep and shallow soils was approximate. Soil depths were mapped at the mapping points, which are often also slope cuttings. For description the limits <0.1m, 0.1 – 0.5m, >0.5m and > 1m were used. Soil depths larger than 1m were an indication of large storage, and only if a large areal extent could clearly be identified (e.g., by evident debris), it was used as additional information to confirm the extent and classification of the polygon. We will add this information for clarification.

From soil physical considerations it can be assumed that a higher fraction of fine material increases storage capacity and decreases permeability (related to soil physical parameters such as field capacity and hydraulic conductivity). 2 mm is often used in soil mapping as a threshold for fine and coarse material (e.g. in the mapping guide of AG BODEN (1994) used in Germany and Austria). For example, soils with a very large fine material fraction (> 80%, corresponding to "loam") were directly classified into low infiltration capacity (GIN). We will give more information on this in the manuscript.

> AG BODEN: Bodenkundliche Kartieranleitung. - 4. Auflage; Bundesanstalt für Geowissenschaften und Rohstoffe und geologische Landesämter der Bundesrepublik Deutschland, Hannover, 1994.

A soil map does not exist in this high Alpine region.

Infiltration class see 3d).

*3c) Vegetation. Please provide more information on your choice between dense and sparse. Any references?*

AR: The method of Markart & Kohl (2004) was used here. "Sparse" vegetation is mainly assigned to the typical high alpine grassland above the karstic bed-rock (mostly as a mosaic pattern). Sparse vegetation was assigned to an area not fully covered by vegetation, i.e. the canopy cover was less than about 90 % in terms of ground area. This information will be added in the revised manuscript. ET and interception processes are of minor importance.

*3d) Infiltration capacity. I agree that there is always subjectivity when the field experimentalist propose his perceptual model of a catchment, but in your case you should make effort in such a way that everybody can reproduce your decision tree. This paragraph lacks the methodology on how the*

*authors make the synthesis of the mapped properties. A clear methodology should be provided to obtain the different classes of each polygon.*

AR: As the infiltration capacity was classified in the field by the field experimentalist based on the combination of the different indicators and by the use of soft data and expert knowledge, there is no *a priori* decision tree. We gave two examples and the typical scale of the polygon lumping in our description to clarify the method. We will add the sequence of the mapping process in more detail as follows: After the "reading the landscape" step the field experimentalist reached the area of interest. Note that, in some cases with limited sight of line, the borders of the homogeneous area (polygon) could no longer be identified. The location of the field experimentalist was checked by GPS. Next, the field experimentalist confirmed the vegetation type and density and its relationship with other variables. Dense vegetation, for example, makes high infiltration capacity (HIN) less likely. Next he/she mapped the soils/debris/loose sediment and looked for appropriate and representative points or slope cuttings where he/she could map the soil depth and check the fine material fraction by sieving (> 80% is directly classified to GIN). Based on the soil type so identified, the spatial homogeneity was carefully checked for extrapolation to the mapped area. Only if the spatial extent of large soil depths could be clear identified, it was used for further classification of the polygon.

Most of the mapped properties so far are related to a certain underlying lithology, so the occurring geologic type (what the field experimentalist could see in the field) must be consistent with this, so was cross-checked with the hydrogeological map. Finally, any traces of surface runoff or erosion visible from a distance, the apparent existence of permanent flow anywhere in the polygon, and any temporary flow or very rare flow (e.g. due to infiltration excess) were used to verify the assessment. The resulting infiltration class is a synthesis of the properties above, i.e. the choice was made to be consistent with the various, complementary pieces of evidence.

*Figure 2 and 5: add scale and orientation of the view*

AR: A note will be added in the figure caption on the extent of, e.g., one of the polygons in the front in Fig. 2 and the length of the main flow path in Fig. 5. North arrows will be added.

*Figure 3 : please make effort to have a real geological cross-section, bedding and fault are missing and the vertical limits between Dolomite, Calcareous Sandstone and Limestone units seems not natural to me.*

AR: The cross section is indeed schematic. We will examine whether there is sufficient field information to add the information above to the cross section, but at this point this does not seem to be the case.

*7) Discussion. Provide reference on the assertion that surface runoff propensity is at the pixel scale (10 m\* 10 m). Discuss not only the advantage of the methodology but also the limits (like subjectivity, compilation of knowledge in different disciplines) and how can you overcome the limits. You should also go beyond the case of study, explaining how field experimentalist are still essential into the understanding of catchment behavior, showing that even if there is subjectivity in their assessment, the role of the field experimentalist is crucial to the hydrological community in the understanding of hydrological basin and their prioposition of perceptual models.*

AR: The working grid size will be justified more fully in chapter 6 in terms of the processes captured and those missed. For the index we chose 10 m x 10 m as a compromise to represent the mapped polygons (smallest polygon A=450 m²) and computational costs. For typical HRUs in catchment modelling the grid size is usually larger, resulting in more pixels with mixed properties, e.g., geology, soils, and the associated infiltration capacity. We will more fully discuss not only the advantage of the methodology but also the limits in the revised manuscript.

We will note the important role of experimentalists, extend the reference to subjectivity, and make further comparisons with other disciplines relying on mapping (vegetation, geology). Of course, the crucial part of the mapping and the translation into hydrological process descriptions is an important part of the message.

*8) Conclusion. Please provide some widening ideas on your method. As example, in flat areas the field person can be help by the use of drones, consequently the method can also be apply in other non-mountainous context. Provide some advice on how the method can be adapt in non-karstic context.*

AR: Aerial photos such as from drones could be useful to resolve more spatial detail than the aerial photos used here, but visibility in the forests is an issue. We will add advice on how to transfer the method to non-karstic catchments as other properties and processes may gain more importance, e.g., lateral subsurface flow, shallow groundwater level fluctuations, connectivity of flow paths, runoff routing, etc.) and, briefly, how these could be captured.

---

## Author Comment (AC3) · 11 Aug 2018

Author's response (AR) to reviewer #3:

We would like to thank the reviewer for his/her insightful comments.

*General comments*

*RC: A "Technical Note" format might proof more useful.*

> AR: We consider this manuscript to go beyond what is usually reported in a technical note. It is not on a technical aspect of an existing method, but broader. We developed and tested the method for a case study area. We therefore believe that a regular paper would be more appropriate for this manuscript.

*RC: The discussion section could strongly improve from discussing limitations of the proposed method more in detail. It could also address whether this method is transferable to other geological settings and catchment types.*

> AR: We will discuss the limitations of the method in more detail as well as the transferability to other (e.g. in non-karstic) regions and additional processes in the revised manuscript.

*Specific Comments*

*RC: p1, l10: "spatial distribution" or "occurrence"*

> AR: It is the spatial distribution of the frequency of runoff occurrence. We will extend the sentence to read ".. to understand the spatial distribution of the frequency of surface runoff ..".

*RC: p1, l17: please put "in the study area" at the end or beginning of this sentence.*

> AR: "in the study region" will be moved to end of sentence.

*RC: P3, l15 How did you delineate the catchment extent? I assume that this is derived from GIS delineation based on the surface topography (not accounting for the subsurface catchment)?*

> AR: There are catchment boundaries available from previous studies estimated by geological and hydrological considerations including isotope data and tracer experiment data (see references). However, the mapped area is somewhat larger than the catchment area, and the plotted polygon follows the mapped area. We will make a note to this effect in the manuscript.

*RC: P4, l16 How was the lookout point chosen? Were there logistical constrains?*

> AR: We will add a short description of how the lookout points were chosen in terms of line of sight, interesting landscape features and logistical constraints.

*RC: P5, l1-5 Please rephrase here*

> AR: We will rephrase adding more detail for clarity.

*RC: P5, l19 show*

> AR: OK

*RC: P8, l11-12 rephrase*

AR: We will rephrase adding more detail for clarity.

*RC: P11, l8 runoff instead of runon*

AR: We actually are meaning to say "runon" to refer to surface runoff that eventually infiltrates. The word "runon" is used in the karst literature and other hydrological literature (e.g., Calvo-Cases et al. 2003).

*Figures*

*RC: Fig. 1 Please put the label "LKAS2" in the map and show the state Austria in a smaller map with the location of this study area.*

AR: OK

*RC: Fig. 2 Why does Zone no 7 not go further downslope into Zone no 1? As the visible bedrock outcrop emerges below Polygon.*

AR: The slope is flatter, so that a thin soil layer has developed in Zone no. 7 resulting in denser vegetation. In No. 1 vegetation is negligible and so it is classified as bare rock. However, one cannot compare this kind of vegetation with the vegetation on e.g., areas with organic soils in no. 17. We will add a statement for clarification.

*RC: Fig. 3 Please enlarge labels in the cross-section.*

AR: Ok

*RC: Fig6/7 Please use a) and b) labels for both subplots. Also use a colour for the Spitzboden catchment extent in the lower subplot.*

AR: OK

---

## Author Response (AR2)

**Author's response**

We are thankful for the valuable comments to improve our discussion section. We considered the following suggestions by the editor in our revised paper:

The causes and effect of the subjectivity involved into this procedure should be better explained. Maybe there could be a dedicated section, about limitations, which could include such discussion.
 It is unclear how much the subjectivity affects the repeatability of the study. Subjectivity is not a bad thing, but the impossibility to repeat the results by another person is problematic. Therefore, it is important that the decisions, even if subjective, are well motivated.

We restructured the discussion section into the three sub-sections and inserted some points to focus more on the strengths and limitations as well as the subjectivity problem. The three sub-sections are:

- 7.1 Strengths and limitations of the method
- 7.2 Surface runoff generation processes
- 7.3 Transferability of the method

Best regards,

**Christian Reszler**

**A propensity index for surface runoff on a karst plateau**

Christian Reszler1, Jürgen Komma2, Hermann Stadler1, Elmar Strobl3, Günter Blöschl2

1JR-AquaConSol GmbH, Steyrergasse 21, 8010 Graz, Austria

[revised manuscript text omitted]